# Two New Species and a New Record of *Microdochium* from Grasses in Yunnan Province, South-West China

**DOI:** 10.3390/jof8121297

**Published:** 2022-12-14

**Authors:** Ying Gao, Guang-Cong Ren, Dhanushka N. Wanasinghe, Jian-Chu Xu, Antonio Roberto Gomes de Farias, Heng Gui

**Affiliations:** 1Center of Excellence in Fungal Research, Mae Fah Luang University, Chiang Rai 57100, Thailand; 2Center for Mountain Futures, Kunming Institute of Botany, Chinese Academy of Sciences, Honghe 654400, China; 3School of Science, Mae Fah Luang University, Chiang Rai 57100, Thailand; 4Guiyang Nursing Vocational College, Guiyang City 550081, China; 5Department of Economic Plants and Biotechnology, Yunnan Key Laboratory for Wild Plant Resources, Kunming Institute of Botany, Chinese Academy of Sciences, Kunming 650201, China

**Keywords:** two new species, *Ascomycota*, endophytes, multigene phylogeny, morphology, new taxa, taxonomy

## Abstract

*Microdochium* species are frequently reported as phytopathogens on various plants and also as saprobic and soil-inhabiting organisms. As a pathogen, they mainly affect grasses and cereals, causing severe disease in economically valuable crops, resulting in reduced yield and, thus, economic loss. Numerous asexual *Microdochium* species have been described and reported as hyphomycetous. However, the sexual morph is not often found. The main purpose of this study was to describe and illustrate two new species and a new record of *Microdochium* based on morphological characterization and multi-locus phylogenetic analyses. Surveys of both asexual and sexual morph specimens were conducted from March to June 2021 in Yunnan Province, China. Here, we introduce *Microdochium graminearum* and *M. shilinense,* from dead herbaceous stems of grasses and report *M. bolleyi* as an endophyte of *Setaria parviflora* leaves. This study improves the understanding of *Microdochium* species on monocotyledonous flowering plants in East Asia. A summary of the morphological characteristics of the genus and detailed references are provided for use in future research.

## 1. Introduction

*Microdochium* is a genus in Microdochiaceae (*Xylariales*, Sordariomycetes) [1,2]. Researchers have studied species in this genus in various countries [3,4,5,6,7,8,9,10,11,12]. Currently, 42 *Microdochium* species are listed in Species Fungorum (http://www.indexfungorum.org/, accessed on 9 September 2022) [13]. However, *Microdochium chuxiongense*, *M. indocalami*, *M. maculosum*, *M. ratticaudae*, *M. salmonicolor*, and *M. yunnanense* were recently introduced [9,10,12,14,15] and, therefore, the total number of species in the genus should be 48.

*Microdochium* species have been collected worldwide, with more frequent collections in Europe and Asia. Where China stands out with the largest number of described species. They are frequently reported as phytopathogens [12], especially in grasses and cereals, causing severe diseases in economically valuable crops. *Microdochium majus* and *M. nivale* cause *microdochium*-patch (also known as pink snow mould or *Fusarium* patch) in wheat and barley [4,16,17,18] and *M. albescens* causes rice leaf-scald [16], with a significant reduction in the crop yield. Tar spot disease, scald disease, root necrosis, and decay of grasses have been reported to be caused by species of *Microdochium* [11]. They have also been reported as saprobes on dead plants [4,19,20,21,22] and as inhabiting rhizosphere soils [4,23] and some species have been reported as endophytes [24,25]. Moreover, Liu et al. [26] isolated *M. lycopodinum* and *M. phragmitis* from aquatic (marine) environments and salmon eggs.

*Microdochium* has also been reported as beneficial to humans. Bioactive compounds of *Microdochium* species can be used against plant pathogens (i.e., *Verticillium dahlia*) [27]. Cyclosporine A, a bioactive compound that has the potential to control human and animal diseases, was isolated from *M. nivale* [28] and extracts of *M. phragmitis* were cytotoxic against human tumoral cell lines [29]. Thus, the biotechnological potential of *Microdochium* species should be explored from natural matrices and preserved for future research [30].

Hyde et al. [1] showed that the descriptive curve had not flattened, while Bhunjun et al. [31] showed that even speciose genera had many more new taxa to be described. In this study, we introduce two novel species, *M. graminearum*, *M. shilinense*, and a new record for *M. bolleyi*, isolated from grasslands in Kunming. This study had the following objectives: (1) to update the phylogenetic analysis of multigene sequence and refine the morphological characters of the genus and (2) to characterize these diverse isolates by incorporating morphological characteristics and molecular data. *Microdochium* species are either very important plant pathogens or non-pathogenic. This study provides information for future research on *Microdochium* and shows it is likely that many novel taxa are yet to be described.

## 2. Materials and Methods

### 2.1. Sample Collection, Isolation, and Identification

Litter and living grass samples were collected from Kunming, Yunnan Province, China, and brought to the laboratory for analysis. Specimens were examined using an Olympus SZ-61 dissecting microscope. Fungal fruiting structures were manually sectioned and mounted in water on a slide to observe their microscopic features. Pure cultures were obtained from litter samples via single spore isolation [32] and from living specimens by the tissue culture isolation method. In brief, leaf blades were cut into small pieces no larger than 1 cm in length, rinsed in sterile distilled water (SDW), and surface-sterilized with 75% ethanol for 3 min, 2.5% NaOCl solution for 0.5–5 min, rinsed in fresh SDW [12,33], blot-dried with sterile paper towels and, finally, cultured in potato dextrose agar (PDA) medium to obtain pure fungi [34]. Micro-morphological characteristics were examined using a Nikon ECLIPSE Ni compound microscope and photographed using a Canon EOS 600D digital camera fitted to the microscope. Photo plates were processed using Adobe Photoshop CS6 Extended version 13.0.1 (Adobe Systems, San Jose, CA, USA), and measurements of morphological structures were processed following the method described in Ren et al. [35]. The living cultures were deposited in the China General Microbiological Culture Collection Center (CGMCC), and the herbaria specimens were deposited in the herbarium of the Kunming Institute of Botany Academia Sinica (HKAS). The new taxa were registered in the Faces of Fungi [36], the Index Fungorum database (http://www.indexfungorum.org/, accessed on 9 September 2022) [13] and the database Fungi of the Greater Mekong Subregion (GMS Microfungi) [37].

### 2.2. DNA Extraction, PCR Amplification, and DNA Sequencing

Genomic DNA was extracted from 50 to 100 mg of axenic mycelium scraped from the edges of the culture grown on PDA at 28 °C for two weeks [38] using the Biospin Fungus Genomic DNA Extraction Kit (BioFlux^®^, Hangzhou, China) following the manufacturer’s protocol. Polymerase chain reaction (PCR) amplifications were carried out for the partial 28S large subunit nuclear ribosomal DNA (LSU), internal transcribed spacer region with intervening 5.8S nrRNA gene (ITS), partial beta-tubulin *tub2*, and partial RNA polymerase II second largest subunit (*rpb2*). The thermal conditions included initial denaturation at 94 °C for 3 min, followed by 35 cycles of denaturation at 94 °C for 10 s, annealing temperatures listed in Table 1, elongation at 72 °C for 20 s, and final extension at 72 °C for 10 min. The total volume of PCR mixtures for amplification was 25 μL containing 8.5 μL ddH_2_O, 12.5 μL 2xF8 FastLong PCR MasterMix (Beijing Aidlab Biotechnologies Co., Ltd., Beijing, China), 2 μL of DNA template, and 1 μL of each forward and reverse primers (stock of 10 pM).

### 2.3. Phylogenetic Analyses

Representative *Microdochium* species used in the phylogenetic analyses were selected from recent studies [7,8,9,10,11,12] and the sequences downloaded from GenBank (https://www.ncbi.nlm.nih.gov/genbank/ (accessed on 30 August 2022)) (Table 2). Individual alignments of LSU, ITS, *tub2*, and *rpb2* sequences were aligned using MAFFT v. 7.475 [43], with default configurations, and trimmed with TrimAl v. 1.3 [44] via the web server Phylemon2 (http://phylemon.bioinfo.cipf.es/utilities.html (accessed on 31 August 2022)). Individual datasets were concatenated into a combined dataset using BioEdit v. 7.0.5.3 [45]. The individual and combined datasets were subjected to maximum likelihood (ML) and Bayesian (BI) phylogenetic inference.

Maximum-likelihood (ML) analysis was performed using RaxML-HPC2 on XSEDE v. 8.2.10 [46] in CIPRES Science Gateway online platform [47], under the GTR+GAMMA model of nucleotide substitution, with 1000 bootstrapping replicates. The evolutionary model of nucleotide substitution for BI was selected independently for each locus using MrModeltest 2.3 [48]. Bayesian inference was conducted by MrBayes on XSEDE v. 3.2.7a in the CIPRES Science Gateway v. 3.3 [47], set with two runs and six simultaneous Markov chain Monte Carlo sampling (MCMC) chains for 2,000,000 generations, and the trees were sampled every 100th generation, for calculating the Bayesian posterior probabilities (BYPP). The first 25% of trees were considered burn-in and discarded. The MCMC heated chain “temperature” was set to the value of 0.15, and the run was stopped automatically when the average standard deviation of split frequencies reached 0.01.

Tree topologies generated in this study were visualized on FigTree v. 1.4.2 [49]. The phylogram was edited in Microsoft Office PowerPoint 2016 (Microsoft Inc., Redmond, WA, USA) and Adobe Photoshop CS6 Extended version 13.0.1 (Adobe Systems, San Jose, CA, USA). New sequences generated from the present study are deposited in GenBank (Table 2).

## 3. Results

### 3.1. Phylogenetic Analyses

The combined sequence data of LSU, ITS, *tub*2, and *rpb*2 consisted of 75 strains of *Microdochium*, *I. lunata* (CBS 204.56), and the newly obtained isolates. A total of 3002 characters, including gaps, were obtained in the phylogenetic analysis, viz. LSU = 1–834, ITS = 835–1394, *tub*2 = 1395–2163, and *rpb*2 = 2164–3002. Phylogenetic analyses obtained from ML and BI methods also resulted in similar topologies.

The seven strains studied here represented three distinct clades (Figure 1). The strains CGMCC 3.23527, CGMCC 3.23528, CGMCC 3.23529, and CGMCC 3.23530 were monophyletic with *M. bolleyi* (CBS 540.92). *Microdochium graminearum* (CGMCC 3.23524 and CGMCC 3.23525) was closely related to *M. seminicola* with support values of 83% ML bootstrap and 1.00 BYPP (Figure 1). *Microdochium shilinense* (CGMCC 3.23531) nested as the basal lineage of the clade containing *M. seminicola, M. graminearum*, and *M. albescens* with strong ML bootstrap (100%) and BYPP (1.00) supports. 

### 3.2. Taxonomy

***Microdochium graminearum*** Y. Gao & H. Gui, *sp. nov*. (Figure 2)

*Index Fungorum number*: IF553518; Facesoffungi number: FoF12703

*Etymology*: The epithet refers to *Gramineae*.

*Holotype*: HKAS 123200

Appear as black spots on a dead herbaceous stem of grasses, visible as black circular or ellipsoid spots on the host surface. ***Sexual morph***: *Ascomata* 100–120 μm diameter × 70–90 μm high (x = 105 × 78 μm, n = 12), scattered, gregarious, deeply immersed in host tissues, subglobose, or elliptical, dark brown to black, uni-loculate, glabrous, non-ostiolate. *Peridium* 5–15 μm thick (x = 9 μm, n = 20), composed of 2–3 layers of flattened, light to dark brown, pseudoparenchymatous cells of *textura angularis*. *Paraphyses* 4–7 μm wide (x = 4.8 μm, n = 20), straight, septate, hyaline, unbranched, broader at the base, tapering towards the apex. *Asci* (55–) 58–73 (–77.6) × (9.6–) 10.6–14.6 (–15.5) μm (x = 65.5 × 12.6 μm, SD =7.7 × 2 μm, n = 20), 8-spored, arising from the base, fusiform, with a short pedicel, bitunicate, hyaline, with a refractive ring around cytoplasmic protrusion, funnel-shaped apical ring. *Ascospores* (16.5–) 18.4–22.5 (–24) × (4–) 4.1–5 (–5.6) μm (x = 20.4 × 4.6 μm, SD = 2 × 0.5 μm, n = 30), slightly overlapping, 1–2-seriate, hyaline, guttulate, lunate, or allantoid to fusiform, with 0–3 transverse septa, often slightly constricted at the medium septum, rounded to slightly pointed at both ends. ***Asexual morph***: Undetermined.

***Culture characteristics*:** Ascospores germinating on PDA within 20 h at room temperature. Germ tube initially produced from the middle ascospore cell. Colonies on PDA reaching 40 mm diameter after four weeks at 20–27 °C, circular, slightly raised, floccose, white from above and yellowish from below, smooth with filamentous edge, mycelium immersed in PDA and grows towards the edge.

***Material examined*:** China, Yunnan Province, Kunming (25°8′19″ N, 102°44′25″ E), on decaying herbaceous grass stem, 20 June 2021, Ying Gao (HKAS 123200, holotype), ex-type culture, CGMCC 3.23525. *ibid.* (HKAS 123199, paratype), ex-paratype culture CGMCC 3.23524.

***Microdochium shilinense*** Y. Gao & H. Gui, ***sp. nov.*** (Figure 3)

*Index Fungorum number*: IF553309; Facesoffungi number: FoF12704

*Etymology*: Named refers to the location (Shilin Yi Autonomous County, China) from where the holotype was collected.

*Holotype*: HKAS 123198

*Saprobic* on a dead herbaceous stem of grass. ***Sexual morph***: *Ascomata* 125–150 μm diameter × 100–120 μm high, (x = 134 × 111 μm, n = 10), scattered, gregarious, deeply immersed in host tissues, globular or subglobose, light brown to black, uni-loculate, non-ostiolate, slightly raised top. *Peridium* 10–20 μm thick (x = 12 μm, n = 30), composed of 3–4 layers of flattened, thick-walled, light brown to dark brown cells of *textura angularis*. *Paraphyses* 3–4.5 μm wide, (x = 3.7 μm, n = 20), straight or curved, septate, hyaline, unbranched, with large to small guttules, slightly constricted at the septa, filiform to stripy. *Asci* (50–) 52–67 (–76) × (7–) 8–9.6 (–10) μm (x = 60 × 8.8 μm, SD = 7 × 1 μm, n = 20), 8-spored, arising from the base, cylindrical, bitunicate, with a short pedicel, hyaline, with refractive ring around cytoplasmic protrusion. *Ascospores* (14–) 15–17 (–18) × (3–) 3.7–4.8 (–5.7) μm (x = 16 × 4.2 μm, SD = 1 × 0.5 μm, n = 30), overlapping, 2-seriate, hyaline, guttulate, fusiform, straight, or curved, with 0–3 transverse septa, sometimes slightly constricted at the medium septum, rounded to slightly pointed at both ends. ***Asexual morph***: Undetermined.

***Culture characteristics*:** Ascospores germinated on PDA within 24 h at room temperature. Germ tube initially produced from the middle cell of the ascospore. Colonies on PDA reaching 50 mm diameter after four weeks at 25–27 °C, circular, slightly raised, smooth, fimbriate, filiform, floccose, white from above and yellowish from below.

***Material examined*:** China, Yunnan Province, Kunming, Shilin Country (24°49′23″ N, 103°32′11″ E), on decaying herbaceous stem of grass, 13 June 2021, Ying Gao (HKAS 123198, holotype), ex-type culture CGMCC 3.23531.

***Microdochium bolleyi*** (R. Sprague) de Hoog & Herm.-Nijh., 1977 (Figure 4)

*Index Fungorum number:* IF 317661; Facesoffungi number: FoF 12706

*Saprobic* on decaying leaves of grass. ***Sexual morph***: Undetermined. ***Asexual morph***: *Mycelium* superficial, consisting of hyaline, finely verruculose, smooth, branched, septate, 1.5–3 µm wide hyphae. *Chlamydospores* 6–8.5 μm diameter, thick-walled, subglobose or ovoid, constricted at the center, hyaline, granulate, terminal, or intercalary, more frequently arranged in chains than clusters. *Conidiogenous cells* cylindrical or oblong, tapering towards both ends, hyaline, smooth, 0–1-septate, (12–) 12.7–14.3 (–14.6) × (3–) 3.3–4 (–4.3) μm (x = 13.5 × 3.6 μm, SD = 0.8 × 0.3 μm, n = 15). *Conidia* aseptate, (6–) 6.6–9 (–10) × (2.3–) 2.5–3.2 (–3.8) μm (x = 7.7 × 2.8 μm, SD = 1 × 3 μm, n = 30), subcylindrical, ellipsoid, or lunate, aseptate, hyaline, smooth-walled, straight, or curved with obtuse apex.

***Culture characteristics*:** Colonies on PDA 50–60 mm in diameter after 15 days at room temperature, mycelia circular, flat, dense, the edges are filamentous and white, grey at center, aerial mycelia cottony or sparse, reverse white. 

***Material examined*:** China, Yunnan Province, Kunming, Kunming Botanical Garden (25°8′19″ N, 102°44′25″ E), on healthy leaves of *Setaria parviflora*, 8 March 2021, Ying Gao (HKAS 123195 paratype), ex-paratype culture CGMCC 3.23528; HKAS 123194, living culture CGMCC 3.23527; HKAS 123196, living culture CGMCC 3.23529; HKAS 123197, living culture CGMCC 3.23530.

## 4. Discussion

Grasses represent the plant family *Poaceae* and include over 10,000 species as herbaceous annuals, biennials, or perennial flowering plants [50]. They play a crucial role in ecosystem functions such as undergrowth, weeds, or as the first members of food cycles [51]. Microfungi can occur on grasses as pathogens, endophytes, epiphytes, or saprobes. In many cases the anamorphs of these microfungi are reported as pathogenic on economically important grasses. Various authors have studied microfungi on grasses [50], and these studies indicated that they have a great diversity; however, there is a lack of information, especially from the Asian region. Therefore, it is important to collect microfungi on grasses in unexploited areas such as Yunnan province in China and assess their taxonomic placements, enabled by both morphological and molecular analyses. In the current study, we describe and illustrate two new species and one new record of microfungi on grasses, viz. *Microdochium graminearum* sp. nov., *M*. *shilinense* sp. nov., and *M*. *bolleyi* from Kunming, Yunnan, based on a biphasic approach (morphological plus molecular analyses) (Figure 1, Figure 2, Figure 3 and Figure 4). *Microdochium graminearum* and *M*. *shilinense* are introduced with their sexual characteristics, whereas *M*. *bolleyi* is accounted for with its asexual morphological features.

*Microdochium graminearum* (HKAS 123200 and HKAS 123199) is introduced as a new species based on its distinct morphology and analysis of a combined LSU, ITS, *tub*2, and *rpb*2 dataset. *M. graminearum* clusters close to *M. seminicola* with 83% ML bootstrap and 1.00 BYPP support (Figure 1). The pairwise nucleotide comparison showed that *M. graminearum* differs from *M. seminicola* (CBS 122706) in 9/550 bp of ITS (1.64%) and 15/860 bp of *rpb*2 (1.74%). Morphologically, the new species differs from *M. seminicola* by its asci and ascospore characteristics. Asci of *M. graminearum* are wider than those of *M. seminicola* (55–77.6 × 9.6–15.5 vs. 41–66.5 × 7.5–11 μm). *M. graminearum* has guttulated ascospores with a rough surface, and *M. seminicola* has smooth-walled ascospores without guttules. Therefore, *M. graminearum* is introduced as a novel taxon based on phylogeny and morphological comparison.

The present phylogenetic analysis showed that *M. shilinense* forms a distinct branch as the basal clade of *M. seminicola*, *M. graminearum,* and *M. albescens* with high bootstrap support (100% ML and 1.00 BYPP) (Figure 1). The pairwise nucleotide comparison showed that *M. shilinense* differs from *M. albescens* (CBS 243.83) in 42/553 bp of ITS (7.59%) and 47/768 bp of *tub*2 (6.12%). *Microdochium shilinense* differs from *M. seminicola* and *M. graminearum* in having cylindrical asci with a refractive ring around cytoplasmic protrusions, while *M. seminicola* has fusiform asci with a funnel-shaped apical ring; *M. graminearum* has fusiform and comparatively larger asci (55–77.6 × 9.6–15.5 vs. 50– 76 × 7–10 μm). *Microdochium shilinense* differs from *M. albescens* in having fusiform ascospores with 0–3 transverse septa, while *M. albescens* has fusoid ascospores with 1–5 transverse septa. Therefore, we introduce *M. shilinense* as a novel taxon.

Phylogeny of a concatenated LSU-ITS-*tub*2-*rpb*2 sequence dataset depicts our *M. bolleyi* isolates as a monophyletic group (Figure 1). Morphologically, our specimens also have hyaline, smooth conidiogenous cells, and aseptate, hyaline, or ellipsoid conidia [23]. However, they differ slightly from CBS 540.92 in having cylindrical conidiogenous cells (12–14.6 × 3–4.3 μm) instead of globose or subglobose conidiogenous cells (2–4.5 × 2–3.5 μm), and larger conidia (6–10 × 2.3–3.8 vs. 5.5–8.5 × 1.6–2.2 μm) [23]. The pairwise nucleotide comparison showed that the new *M. bolleyi* isolates differ from the CBS 540.92 *M. bolleyi* in 1/832 bp of LSU (0.12%), 2/543 bp of ITS (0.36%), 16/840 bp of *rpb*2 (1.90%), and 9/770 bp of *tub*2 (1.17%). Therefore, we introduced *M. bolleyi* as a new host and country record from *Setaria parviflora* leaves in China.

## 5. Conclusions

In conclusion, we isolated seven fungi associated with *Microdochium* on grasses by single spore and tissue isolations. Based on morphology and phylogeny, they were identified as *Microdochium graminearum* sp. nov., *M*. *shilinense* sp. nov., and *M*. *bolleyi*. As many *Microdochium* species have been reported from China (Table 3), we believe that abundant *Microdochium* species will be discovered in future studies. Our results also highlight that Yunnan Province has not yet been properly studied and is an open field for new fungal discoveries.

## Figures and Tables

**Figure 1 jof-08-01297-f001:**
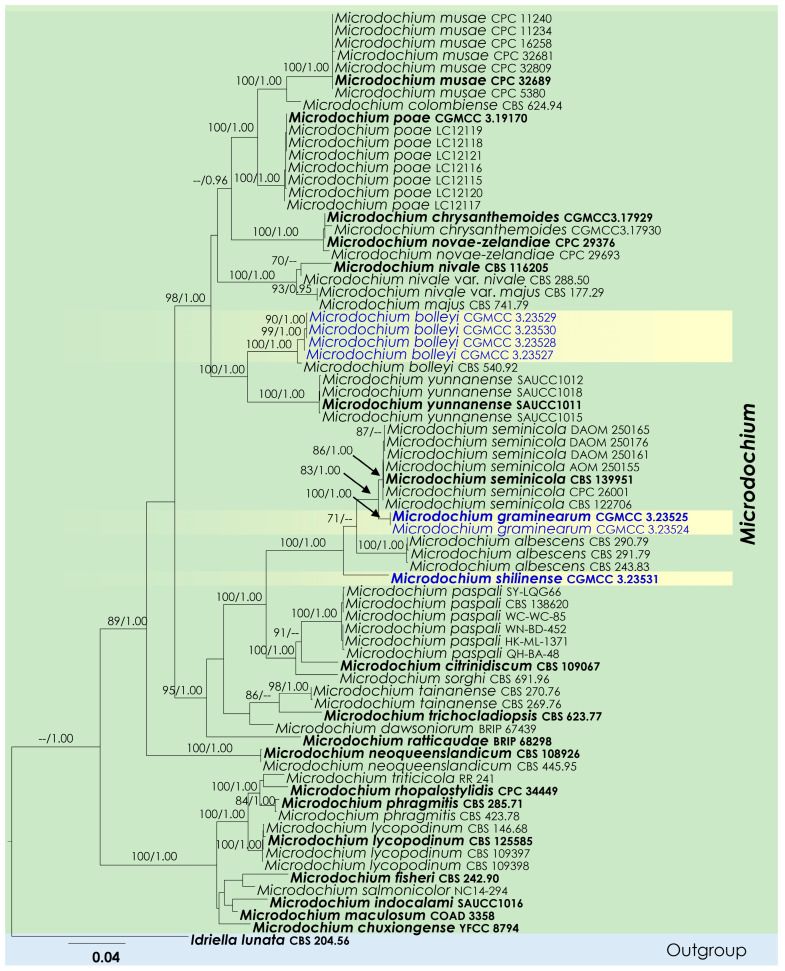
Phylogenetic tree of *Microdochium* species based on maximum likelihood analysis of a combined multigene alignment (LSU, ITS, *tub2*, and *rpb2*). Bootstrap support values for ML higher than 70% and Bayesian posterior probabilities (PP) higher than 0.95 are indicated at the node. *Idriella lunata* (CBS 204.56) was used as the outgroup. Ex-type strains are in bold font; the newly generated sequences are denoted in blue.

**Figure 2 jof-08-01297-f002:**
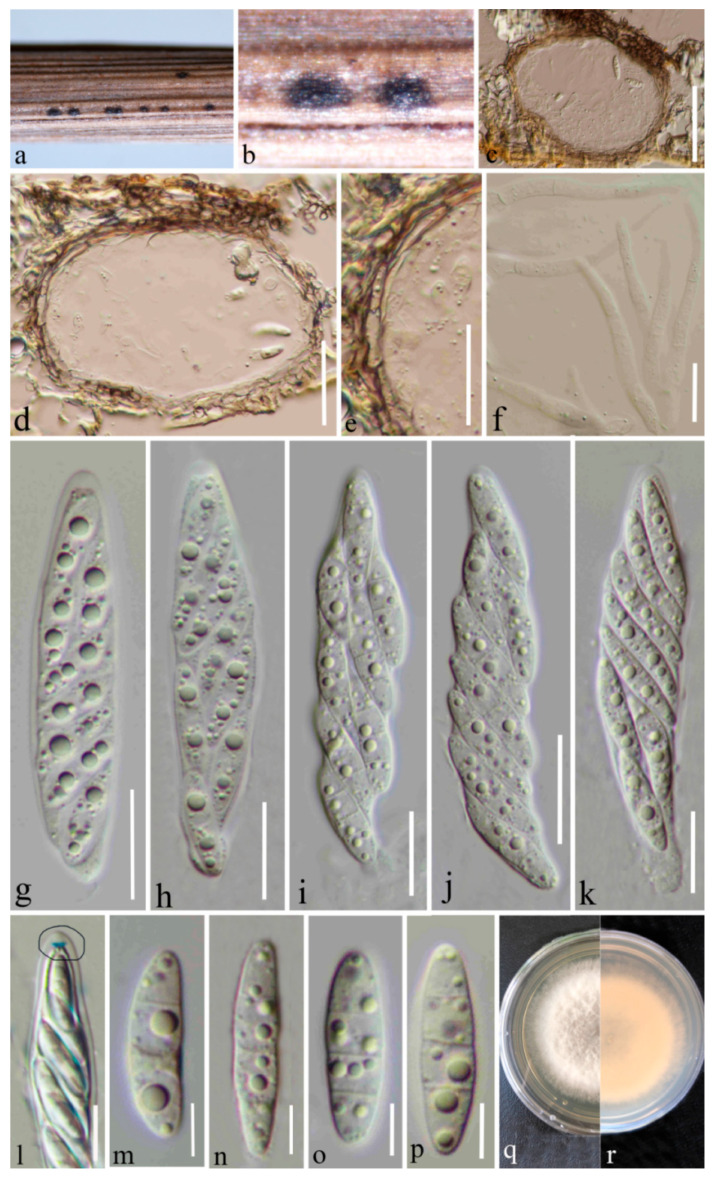
*Microdochium graminearum* (HKAS 123200, holotype). (**a**,**b**) Appearance of immersed ascomata on the host; (**c**,**d**) vertical section of the ascoma; (**e**) peridium; (**f**) paraphyses; (**g**–**k**) asci; (**l**) asci stained by Melzer’s reagent, showing a refractive ring around cytoplasmic protrusion (black circle); (**m**–**p**) ascospores; (**q**) surface of colony on PDA; and (**r**) reverse of colony on PDA. Scale bars (**c**) 50 μm; (**d**) 30 μm; (**e**,**f**) 20 μm; (**g**–**k**) 15 μm; (**l**) 10 μm; and (**m**–**p**) 5 μm.

**Figure 3 jof-08-01297-f003:**
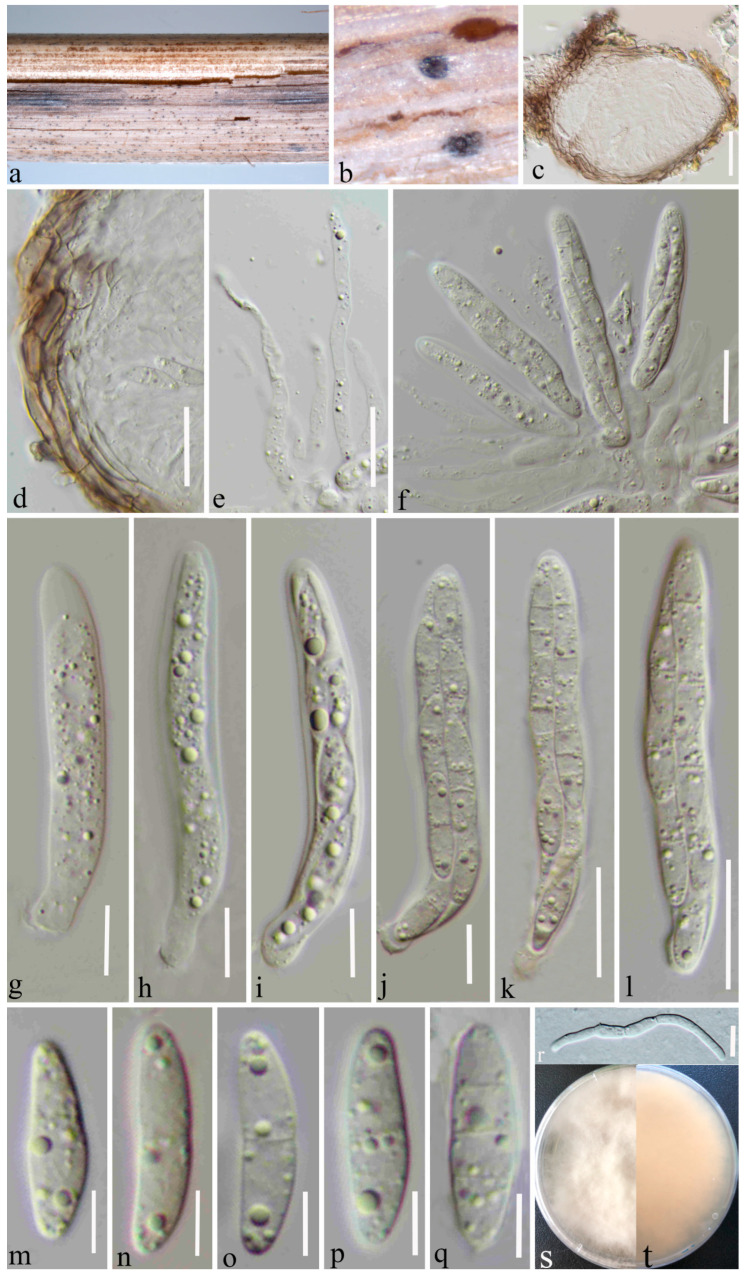
*Microdochium shilinense* (HKAS 123198, holotype). (**a**,**b**) Appearance of immersed ascomata on the ©t; (**c**) vertical section of the ascoma; (**d**) peridium. (**e**) paraphyses; (**f**–**l**) asci; (**m**–**q**) ascospores; (**r**) germinated ascospores; (**s**) surface of the colony on PDA; and (**t**) reverse of the colony on PDA. Scale bars, (**c**–**f**) 20 μm; (**g**–**j**) 10 μm; (**k**,**l**) 20 μm; and (**m**–**r**) 10 μm.

**Figure 4 jof-08-01297-f004:**
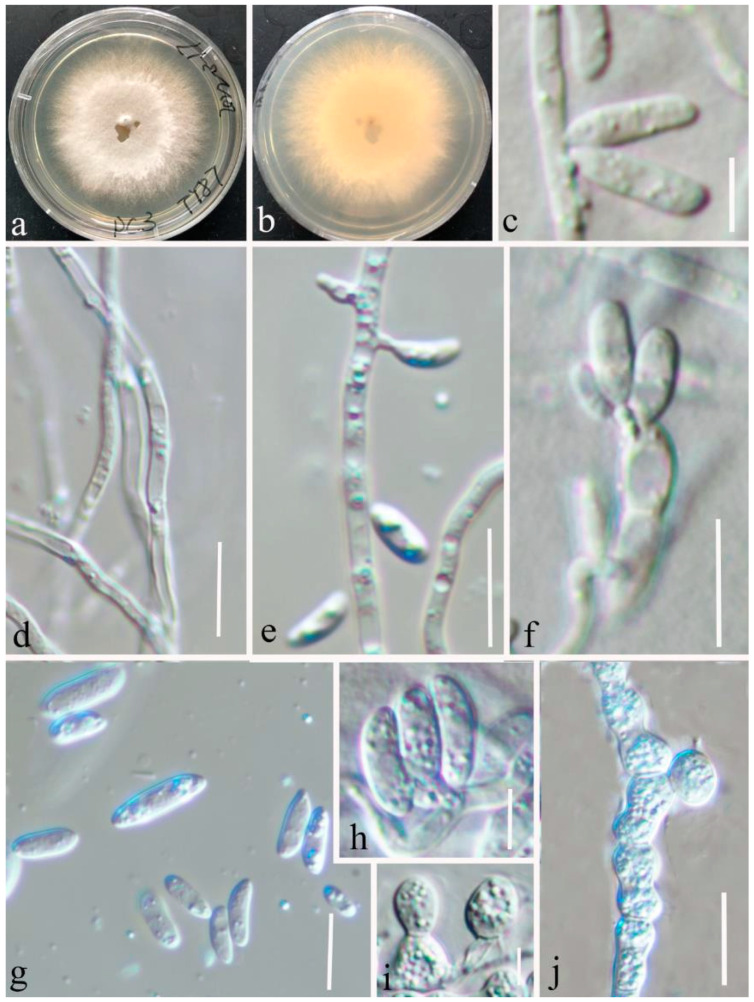
*Microdochium bolleyi* (HKAS 123195) on leaves of healthy *Setaria parviflora*. (**a**) The surface of the colony on PDA; (**b**) the reverse of the colony on PDA; (**d**) hyaline mycelium; (**c**,**e**,**f**) conidiophores and conidiogenous cells; (**g**,**h**) conidia; and (**i**,**j**) chlamydospores. Scale bars, (**c**) 5 μm; (**d**–**g**) 10 μm; (**h**,**i**) 5 μm; and (**j**) 15 μm.

**Table 1 jof-08-01297-t001:** Polymerase chain reaction (PCR) thermal cycle program for the genetic markers used in this study.

Genes/Loci	PCR Primers (Forward/Reverse)	PCR Annealing Thermal Conditions	References
ITS	ITS5/ITS4	55 °C for 15 s	[39]
LSU	LR0R/LR5		[40]
*tub*2	Btub526F and Btub1332R	55 °C for 30 s	[18]
*rpb2*	fRPB2-5F2/fRPB2-7cR	57 °C for 50 s	[41,42]

**Table 2 jof-08-01297-t002:** GenBank accession numbers of the strains used for phylogenetic analysis in this study.

Species Name	Strain Numbers	GenBank Accession Numbers			
		LSU	ITS	*tub*2	*rpb*2
*Idriella lunata*	CBS 204.56 *	KP858981	KP859044	NA	NA
*Microdochium albescens*	CBS 290.79	KP858950	KP859014	KP859078	KP859123
*M. albescens*	CBS 291.79	KP858932	KP858996	KP859059	KP859105
*M. albescens*	CBS 243.83	KP858930	KP858994	KP859057	KP859103
*M. bolleyi*	CBS 540.92	KP858946	KP859010	KP859073	KP859119
* M. bolleyi *	CGMCC 3.23527	OP104018	OP103968	OP242830	NA
* M. bolleyi *	CGMCC 3.23528	OP104019	OP103969	OP242831	NA
* M. bolleyi *	CGMCC 3.23529	OP104020	OP103970	OP242832	OP184897
* M. bolleyi *	CGMCC 3.23530	OP104021	OP103971	OP242833	OP184898
*M. chrysanthemoides*	LC 5363 *	KU746736	KU746690	NA	NA
*M. chrysanthemoides*	LC 5466	KU746735	KU746689	NA	NA
*M. citrinidiscum*	CBS 109067 *	KP858939	KP859003	KP859066	KP859112
*M. colombiense*	CBS 624.94 *	KP858935	KP858999	KP859062	KP859108
*M. chuxiongense*	YFCC 8794 *	OK586160	OK586161	OK556901	OK584019
*M. dawsoniorum*	BRIP 67439	NA	MN492650	NA	NA
*M. fisheri*	CBS 242.90 *	KP858951	KP859015	KP859079	KP859124
* M. graminearum *	CGMCC 3.23524	OP104015	OP103965	OP242835	OP236026
* M. graminearum *	CGMCC 3.23525 *	OP104016	OP103966	OP236029	OP236027
*M. indocalami*	SAUCC 1016 *	MT199878	MT199884	MT435653	MT510550
*M. lycopodinum*	CBS 125585 *	KP858952	KP859016	KP859080	KP859125
*M. lycopodinum*	CBS 146.68	KP858929	KP858993	KP859056	KP859102
*M. lycopodinum*	CBS 109397	KP858940	KP859004	KP859067	KP859113
*M. lycopodinum*	CBS 109398	KP858941	KP859005	KP859068	KP859114
*M. maculosum*	COAD 3358 *	OK966953	OK966954	NA	NA
*M. majus*	CBS 741.79	KP858937	KP859001	KP859064	KP859110
*M. musae*	CBS 111018	NA	AY293061	NA	NA
*M. musae*	CBS 143499	MH107941	MH107894	NA	NA
*M. musae*	CBS 143500 *	MH107942	MH107895	NA	MH108003
*M. musae*	CPC:11234	MH107943	MH107896	NA	NA
*M. musae*	CPC:11240	MH107944	MH107897	NA	NA
*M. musae*	CPC:16258	MH107945	MH107898	NA	NA
*M. musae*	CPC:32681	MH107946	MH107899	NA	NA
*M. neoqueenslandicum*	CBS 445.95	KP858933	KP858997	KP859060	KP859106
*M. neoqueenslandicum*	CBS 108926 *	KP858938	KP859002	KP859065	KP859111
*M. nivale*	CBS 116205 *	KP858944	KP859008	KP859071	KP859117
*M. nivale var. majus*	CBS 177.29	MH866500	MH855031	NA	NA
*M. nivale var. nivale*	CBS 288.50	MH868135	MH856626	NA	NA
*M. novae-zelandiae*	CPC:29376 *	NG_066396	NR_172274	LT990608	LT990641
*M. novae-zelandiae*	CPC:29693	LT990628	LT990656	LT990609	LT990642
*M. paspali*	CBS 138620 *	NA	NR_158810	NA	NA
*M. paspali*	CBS138620	NA	KJ569509	KJ569514	NA
*M. paspali*	QH-BA-48	NA	KJ569510	KJ569515	NA
*M. paspali*	SY-LQG66	NA	KJ569511	KJ569516	NA
*M. paspali*	WC-WC-85	NA	KJ569512	KJ569517	NA
*M. paspali*	WN-BD-452	NA	KJ569513	KJ569518	NA
*M. phragmitis*	CBS 285.71 *	KP858949	KP859013	KP859077	KP859122
*M. phragmitis*	CBS 423.78	KP858948	KP859012	KP859076	KP859121
*M. poae*	CGMCC 3.19170 *	NA	MH740898	MH740914	MH740906
*M. poae*	LC 12115	NA	MH740901	MH740917	MH740909
*M. poae*	LC 12116	NA	MH740902	MH740918	MH740910
*M. poae*	LC 12117	NA	MH740903	MH740919	MH740911
*M. poae*	LC 12118	NA	MH740897	MH740913	MH740905
*M. poae*	LC 12119	NA	MH740899	MH740915	MH740907
*M. poae*	LC 12120	NA	MH740904	MH740920	MH740912
*M. poae*	LC 12121	NA	MH740900	MH740916	MH740908
*M. ratticaudae*	BRIP 68298 *	MW481666	MW481661	NA	MW626890
*M. rhopalostylidis*	CPC:34449 *	MK442532	MK442592	NA	MK442667
*M. salmonicolor*	NC14-294	MK836108	MK836110	NA	NA
*M. seminicola*	KAS 3576 *	KP858974	KP859038	KP859101	KP859147
*M. seminicola*	KAS 1516	KP858961	KP859025	KP859088	KP859134
*M. seminicola*	KAS 3574	KP858973	KP859037	KP859100	KP859146
*M. seminicola*	KAS 3158	KP858970	KP859034	KP859097	KP859143
*M. seminicola*	KAS 1527	KP858966	KP859030	KP859093	KP859139
*M. seminicola*	KAS 1473	KP858955	KP859019	KP859082	KP859128
*M. seminicola*	CBS 122706	KP858943	KP859007	KP859070	KP859116
* M. shilinense *	CGMCC 3.23531 *	OP104022	OP103972	OP242834	NA
*M. sorghi*	CBS 691.96	KP858936	KP859000	KP859063	KP859109
*M. tainanense*	CBS 269.76 *	KP858945	KP859009	KP859072	KP859118
*M. tainanense*	CBS 270.76	KP858931	KP858995	KP859058	KP859104
*M. trichocladiopsis*	CBS 623.77 *	KP858934	KP858998	KP859061	KP859107
*M. triticicola*	RR 241	NA	AJ748691	NA	NA
*M. yunnanense*	SAUCC 1011 *	MT199875	MT199881	MT435650	MT510547
*M. yunnanense*	SAUCC 1012	MT199876	MT199882	NA	MT510548
*M. yunnanense*	SAUCC 1015	MT199877	MT199883	MT435652	MT510549
*M. yunnanense*	SAUCC 1018	MT199880	MT199886	MT435655	NA

* Denotes ex-type or ex-epitype strains. The newly generated sequences are indicated in blue, NA: not available. Abbreviations: HKAS, Cryptogamic Herbarium of Kunming Institute of Botany, Academia Sinica, Kunming, China; LC, culture collection (personal culture collection held in the laboratory of Dr. Lei Cai); BRIP, Queensland Plant Pathology Herbarium (BRIP); CGMCC, China General Microbiological Culture Collection Center; CPC, culture collection of Pedro Crous housed at CBS; SAUCC, Shandong Agricultural University Culture Collection; CBS, Centraalbureau voor Schimmelcultures, Utrecht, the Netherlands; RR, Rothamsted Research, Harpenden, UK; YFCC, Yunnan Fungal Culture Collection of Yunnan University.

**Table 3 jof-08-01297-t003:** List of *Microdochium* species reported worldwide.

Name of Taxon	Host	Place	Life-Mode	References
*Microdochium albescens*	*Oryza sativa*	Ivory Coast	Plant pathogen	[4]
*M. bolleyi*	*Gramineae*, wood, *Setaria parviflora*	North Dakota, U.S.A.; Syria, Canada; China	Plant pathogen, endophyte	[23,52,53], this study
*M. caespitosum*	Dead leaves	Tanzania	Saprophyte	[21]
*M. chrysanthemoides*	Air of a karst cave	China	―	[5]
*M. citrinidiscum*	Leaf of Eichhornia crassipes	Peru	Pathogen	[4]
*M. colombiense*	*Musa sapientum*	Colombia	―	[4]
*M. chuxiongense*	On pileus of *Bondarzewia* sp.	China	―	[15]
*M. consociatum*	―	San Jorge Province (Ecuador)	―	[4]
*M. cylindricum*	Dead leaves of Eucalyptus	Brazil	Saprophyte	[22]
*M. dawsoniorum*	Leaves of Sporobolus natalensis	Australia	―	[8]
*M. fisheri*	Stem of *Oryzae* sativa, Rhizospheric paddy soil	U.K.; India	Endophyte	[4,30]
*M. fusariisporum*	Dead straw of *Panicum virgatum*	Kansas, U.S.A.	Saprophyte	[4]
*M. graminearum*	Gramineae	China	Saprophyte	This study
*M. griseum*	Dead leaves of Sapium ellipticum	Tanzania	Saprophyte	[21]
*M. indocalami*	Leaves of Indocalamus longiauritus	China	Plant pathogen	[12]
*M. intermedium*	Soi1	Papua New Guinea	―	[23]
*M. linariae*	Stem	Italy	―	[54]
*M. lycopodinum*	*Lycopodium annotinum*, Phragmites australis, air, salmon eggs	Austria; Germany; Netherlands	Non-pathogenic	[4,25,26]
*M. maculosum*	Leaves of Digitaria insularis	Brazil	Plant pathogen	[10]
*M. majus*	On Triticum aestivum	Germany	Plant pathogen	[4,17]
*M. maydis*	Leaves of Zea mays	Mexico	Plant pathogen	[4,55]
*M. musae*	Leaves of Musa sp.	China (Taiwan)	Plant pathogen	[6]
*M. neoqueenslandicum*	*Juncus effusus*, *Agrostis* sp.	Waihi, New Zealand; Netherlands	Plant pathogen	[4]
*M. nivale*	Roots of Triticum aestivum; Porteresia coarctata	UK	Plant pathogen	[28,56]
*M. novae-zelandiae*	Leaves of Poaceae	New Zealand	Plant pathogen	[11]
*M. opuntiae*	Dead leaves of Oputia	Louisiana, U.S.A.; Langlois	Plant pathogen	[4,57]
*M. oryzae*	*Oryzae sativa*	Japan	Plant pathogen	[56]
*M. palmicola*	Dead petiole of Roystonea regia	Cuba	Saprophyte	[19]
*M. panattonianum*	Leaves of Lactuca sativa	Denmark	Plant pathogen	[58]
*M. paspali*	*Paspalum vaginatum*	China (Hainan)	Pathogen	[59]
*M. passiflorae*	Dead stem of Passiflora edulis	New Zealand	Saprophyte	[20]
*M. phragmitis*	*Phragmitis communis*, *Phragmites australis*, salmon eggs, angiosperms	Germany; Poland; Antarctic	Endophyte	[3,26,29]
*M. phyllanthi*	Leaves of Phyllanthus discoideus	Germany; Poland	Plant pathogen	[21]
*M. poae*	Leaves of Poa pratensis and Agrostis stolonifera	China	Plant pathogen	[60]
*M. punctum*	Stem of Sisyrinchii campestris	U.S.A.	―	[61]
*M. queenslandicum*	Forest soil	Australia	―	[62]
*M. ratticaudae*	Stem of Sporobolus natalensis (Poaceae)	Australia	―	[9]
*M. rhopalostylidis*	Leaves of Rhopalostylis sapida	New Zealand	Plant pathogen	[7]
*M. salmonicolor*	Soil	Korea	―	[14]
*M. sclerotiorum*	Culture contaminant	Netherlands	―	[63]
*M. seminicola*	Grain seeds, barley, Triticum aestivum	Canada; Switzerland	Plant pathogen	[4]
*M. shilinense*	Gramineae	China	Saprophyte	This study
*M. sorghi*	Leaves of Sorghum vulgaris	Louisiana, U.S.A.; Cuba	Pathogen	[12,16,52]
*M. stevensonii*	*Panicum hemitomon*	Florida, U.S.A.	―	[4,64]
*M. stoveri*	*Musa* sp.	Honduras, Central America	Plant pathogen	[56]
*M. tainanense*	Root of Saccharum officinarum	Japan; China (Taiwan)	Rhizosphere fungus	[4,23]
*M. trichocladiopsis*	Rhizosphere of Triticum aestivum	Unknown country	Rhizosphere fungus	[4]
*M. triticicola*	Roots of *Triticum aestivum*	UK	Plant pathogen	[65]
*M. yunnanense*	Leaves of Indocalamus longiauritus	China	Plant pathogen	[12]

## Data Availability

All newly generated sequences were deposited in GenBank (https://www.ncbi.nlm.nih.gov/genbank/ (accessed on 16 September 2022), Table 2).

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
