# Peer review of "Two New Species and a New Record of Microdochium from Grasses in Yunnan Province, South-West China"

_jof, 2022, doi:10.3390/jof8121297_

Round 1

Reviewer 1 Report

The main question addressed by the research is the diversity of Microdochium sp. in Yunnan Province, China – Isolation and characterization of diverse Microdochium isolate from samples collected in Yunnan Province, China based on morphological characters and phylogenetic analysis of multi-gene sequence data.  It's an original topic in the field. It addresses a specific gap in Microdochium diversity, resolving taxonomic and nomenclatural uncertainty.  It discovers New Microdochium species and region-dependent geographical plant-fungal diversity patterns. The author should consider adding taxonomic information regarding the plant samples (Genus or species level ID of grass host). The conclusions are consistent with the evidence and arguments presented and they address the main question posed well. The references and figures look fine.

Table 1 shows Microdochium bolleyi on Setaria parviflora is being already reported in other countries. Then why the statement “and a new record (M. bolleyi) on Setaria parviflora” in the conclusion section? Maybe the authors meant to state - a new record (first report) of M. bolleyi on Setaria parviflora in China?

Author Response

Authors are obliged to you for your positive and encouraging comments. Could you please see the updated manuscript.

Reviewer 2 Report

The methods of the article are correct and well described, the work of this fungual genus is interesting specially because of the plant pathogen species. However your article should be imprved, in the formal aspect your article is so extense because of long and detailed tables based not only in your samples and at least partially this should be moved out of the main body of the article.

Should be clearer if you provide at least one small table with only information of your samples, as number of specimens, collection place, host... 

My major concern is that you are claiming M kunmingense as new species. When you are adding only two specimens and they are gouped together, this is not "sister clades" this is in fact a clade. You don´t have evidence that M. kunmingense is different from M. rhopalostylidis. You shoud analyze more samples and compare sexual phase. Without genetic difference and only with the anamorph, I don´t understand how you conlcude that these species are different. Even more, you didn´t explained it in the reslts neither in discussion section. For this reason I am in disagree with the paper.

  Table 1 is so long and is not mainly work done in this study, I suggest move it to supplementary material or appendix.

L 128 cite the author of Figtree (Rambaut) instead of the link to download page.

L 148-149 You didn´t compare the tree topologies using a specific software? How different were the single sequence trees? You should provide it at least as suppelementary material.

Based on the phylogenetic tree, M. kunmingense Is not different of M. rhopalostylidis, for this I don´t agree the recognition of this new species. To my opinion they are conspecific, and you should add more data and new analyses to demonstrate that M. kummingense is a new species.

Witout differecn in the phylogenetic tree (is grouped together with M. rhopalostylidis with 100% and 1 of bootstrap and Bi) and without the sexual morph I consider that you cannot claim a new species from of M. kummingense.

The discussion is poor, doesn´t analyze case by case your new species and should be completely rewritten in order to provide strength to your work.

Conclusions should be also improved.

Table 4 is so extense and should be moved to supplementary material.

Author Response

(The authors gave the same response as above.)

Round 2

Reviewer 2 Report

Most of the suggestions have been applied, other suggestions are almost style.

Avoiding the claiming of the third new species discovered my concerns disminissed so much.